# Legged Robot State Estimation using Invariant Kalman Filtering and Learned Contact Events

**Tzu-Yuan Lin, Ray Zhang, Justin Yu, and Maani Ghaffari**
University of Michigan, Ann Arbor, MI, USA
{tzuyuan,rzh,yujustin,maanigj}@umich.edu

**Abstract:** This work develops a learning-based contact estimator for legged robots that bypasses the need for physical sensors and takes multi-modal proprioceptive sensory data as input. Unlike vision-based state estimators, proprioceptive state estimators are agnostic to perceptually degraded situations such as dark or foggy scenes. While some robots are equipped with dedicated physical sensors to detect necessary contact data for state estimation, some robots do not have dedicated contact sensors, and the addition of such sensors is non-trivial without redesigning the hardware. The trained network can estimate contact events on different terrains. The experiments show that a contact-aided invariant extended Kalman filter can generate accurate odometry trajectories compared to a state-of-the-art visual SLAM system, enabling robust proprioceptive odometry.

**Keywords:** State Estimation, Deep Learning, Legged Robot, Invariant EKF

## 1 Introduction

Legged robots can traverse uneven terrains. This unique capability gives them the potential to conduct scientific exploration in extreme environments, execute rescue missions in hazardous scenes, and can help humans in everyday tasks [1]. To accomplish these goals, knowledge of the robot's current state is necessary. While joint angles can be directly measured using encoders, the robot's pose and velocity require additional measurements and a mathematically sound data fusion method.

Vision-based state estimators might suffer from failure due to illumination changes or extreme environments such as snowstorms or foggy scenes [2]. In contrast, proprioceptive state estimators are agnostic to perceptually degraded situations and can operate at a high update rate (e.g., $100 - 2000\,\text{Hz}$). This high-frequency information provides odometry estimates for localization and mapping tasks [3, 4, 5] and controllers and planners to maintain stability and execute planned policies [6]. The former is the focus of this work.

Proprioceptive state estimators often fuse the measurements from an Inertial Measurement Unit (IMU) with leg odometry. Leg odometry, in particular, uses kinematics and contact data to update the state. Reliable measurements of kinematic and contact information become critical in this case. Not all legged robots are equipped with dedicated contact sensors or springs to detect contact [7, 8]. The addition of dedicated contact sensors is non-trivial and often leads to hardware redesign.

In this paper, we develop a deep learning-based contact estimator that does not require dedicated sensors and instead uses joint encoders, kinematics, and IMU data. We create contact data sets using an MIT Mini Cheetah robot [8] on eight different terrains. We further deploy the contact estimator along with a contact-aided invariant extended Kalman filter (InEKF) [9] and show that the resulting odometry trajectory is comparable to a state-of-the-art visual SLAM algorithm (used as a proxy for ground truth). The contributions of this work are as follows. 1) Open-source contact data sets recorded using an MIT Mini Cheetah; 2) A lightweight learning-based contact estimator that mitigates the need for physical contact sensors for state estimation tasks; 3) A quadruped version of the contact-aided invariant EKF compatible with Lightweight Communications and Marshalling (LCM) [10, 11] interface. The data sets and software are available for download [1]; 4) Experimental

---

[1]https://github.com/UMich-CURLY/deep-contact-estimator
https://github.com/UMich-CURLY/cheetah_inekf_realtime

5th Conference on Robot Learning (CoRL 2021), London, UK.

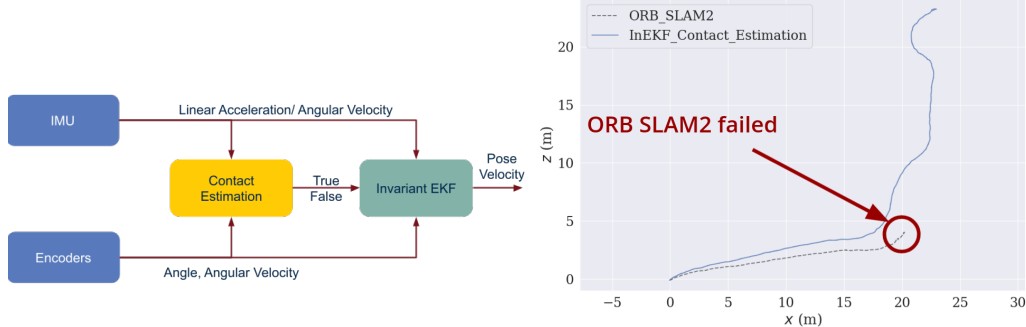

Figure 1: Left: Proposed state estimator. The deep contact estimation network (Yellow) takes joint encoders, IMU, and kinematics data as input and classifies the contact state for the quadruped robot. For state estimation, the estimated contacts, the IMU, and joint encoders data are fused inside a contact-aided invariant extended Kalman filter. Right: The trajectories of the Invariant EKF and ORB SLAM2 in an asphalt and forest data set. The proposed state estimator can serve as reliable odometry when visual SLAM systems fail.

results for contact-aided state estimation using an MIT Mini Cheetah on 8 different terrains. A video demonstrating the proposed method can be found at: https://youtu.be/oVbP-Y8xT_E.

## 2   Related Work

Model-based approaches segment the touchdown event of robot legs or prosthetic legs by thresholding on the estimated Ground Reaction Force (GRF) from the general equation of motion [12, 13, 14]. Although this method can detect touchdown events, the estimated GRF is often noisy and unreliable, especially for lightweight robots. De Luca et al. [15], Haddadin et al. [16] proposed a Generalized Momentum (GM) method for detecting contact events on robot manipulators. This GM-based method is, in fact, a filtered version of the work of Focchi et al. [12]. Although GM-based methods mitigate the noise problem in GRF estimation, an empirical threshold on the cut-off frequency is still required.

Hwangbo et al. [17] introduces a probabilistic representation of the contact state and uses a Hidden Markov Model (HMM) to fuse the dynamics and kinematics for contact estimation. They adopt a Monte-Carlo sampling algorithm to compute the transition model and verify the method against GM-based methods. Jenelten et al. [18] expands the HMM method and focuses on slippage detection. They demonstrate ANYmal [19], a quadruped robot, walking stably on slippery ground. The above two methods aim to detect contact as early as possible for the controller to maintain stability; however, we aim to detect contact intervals for state estimation on various terrains.

Bledt et al. [20] leverages both the GM-based methods and the probabilistic representation of contact states. They use a Kalman filter to fuse the gait phase scheduler information from the controller with the GRF estimated using the GM method and demonstrate that estimates contacts can help the controller reduce the bouncing event upon touchdown. However, this method assumes the leg phase to be periodic as it uses the gait scheduler information in the prediction step of the Kalman filter. It could experience a loss in performance when the phase is heavily violated as the robot interacts with uneven terrains.

Data-driven approaches take advantage of the rapid development of recent machine learning techniques. Camurri et al. [21] uses logistic regression to learn the GRF threshold for contact detection. This work compares against heuristic-based thresholding on GRF using a base state estimator. The result shows that the logistic regression classifier can double the performance of the state estimator. However, compared to deep learning methods, the performance of a logistic regression classifier gets saturated as the number of data increases [22, 23]. This method requires a specific training procedure for different gait, loading conditions of the robot, and individual terrain properties.

Rotella et al. [24] uses a fuzzy C-means clustering for the probability of contacts in all six end-effector degrees of freedom. They integrate the contact estimator with a base state estimator and show their approach performs considerably better than implementations that are purely based on measured normal force. However, this method assumes contact wrench sensors and additional IMU are available at each end-effector. Furthermore, this method was only tested in simulation. Its per-

formance on real robots remains unknown. Piperakis et al. [25] proposes an unsupervised learning method for humanoid gait phase estimation. The authors employ Gaussian Mixture Models (GMMs) for clustering and show the accuracy by comparing to the ground-truth data and leg odometry. However, this work also assumes the availability of wrench/force sensors at each end-effector, and the clustering result is affected by the gait and data density.

The above methods either assume the availability of wrench/force sensors or are restricted by the nature of simple regression and are thus unable to generalize to different scenarios. In contrast, our work proposes a multi-modal deep learning-based contact estimator that does not require contact sensors and can generalize well to different gaits and terrain properties. Moreover, as more data becomes available, the network performance can be improved.

## 3  Preliminaries

**State Representation.** We wish to estimate the orientation, velocity, and position of the body (IMU) frame in the world frame at any time stamp $t$, which are represented as $R_t \in \mathrm{SO}(3)$, $v_t \in \mathbb{R}^3$, and $p_t \in \mathbb{R}^3$, respectively. We define the state $X_t \in \mathrm{SE}_{L+2}(3)$ as $X_t := (R_t, v_t, p_t, d_{lt})$, where $d_{lt} \in \mathbb{R}^3$ is the position of a contact foot in the world frame. When a contact event is detected for leg $l$, $d_{lt}$ is calculated via the forward kinematics and the current body state, and then augmented into the state to enforce the no-slip (zero-velocity) condition. If the contact constraint breaks, the associated contact position $d_{lt}$ will be marginalized from the state to allow foot movement. The matrix Lie group, $\mathrm{SE}_{L+2}(3)$ is an extension of $\mathrm{SE}(3)$ and it was introduced by Barrau [26].

When a new contact event is detected, the position of the corresponding foot is computed using $\bar{d}_t = \bar{p}_t + \bar{R}_t h_p(\tilde{\alpha}_t)$, where $h_p(\tilde{\alpha}_t)$ is the foot position relative to the body frame computed by the forward kinematics (the bar notation denotes estimated variables). $L$ indicates the number of legs that is currently having contact with the ground (here $L \leq 4$).

**Process Model.** We assume the IMU measurements, i.e., angular velocity and linear acceleration, are corrupted by white Gaussian noise:

$$\tilde{\omega}_t = \omega_t + w_t^g, \quad w_t^g \sim \mathcal{GP}(0_{3,1}, \Sigma^g \delta(t - t')),$$
$$\tilde{a}_t = a_t + w_t^a, \quad w_t^a \sim \mathcal{GP}(0_{3,1}, \Sigma^a \delta(t - t')),$$

where $\mathcal{GP}$ represents a Gaussian process and $\delta(t - t')$ is the Dirac delta function. To handle small foot slippage, we model the contact velocity noise in the body frame via a white Gaussian noise $w_t^v \sim \mathcal{GP}(0_{3,1}, \Sigma^v \delta(t - t'))$ [27]. The process model of the individual term in the state becomes:

$$\frac{d}{dt}R_t = R_t(\tilde{\omega}_t - w_t^g)_\times, \quad \frac{d}{dt}v_t = R_t\tilde{a}_t - w_t^a + g,$$
$$\frac{d}{dt}p_t = v_t, \quad \frac{d}{dt}d_t = R_t h_R(\tilde{\alpha}_t)(-w_t^v),$$

where $(\cdot)_\times$ denotes a $3 \times 3$ skew symmetric matrix, $g$ is the gravity vector, $\tilde{\alpha}_t$ is the encoder measurements, and $h_R(\tilde{\alpha}_t)$ is the orientation of the contact frame in IMU (body) frame calculated by forward kinematics using FROST library [28].

**Measurement Model.** We use forward kinematics from current encoder measurements in a right-invariant model described by Hartley et al. [9]. We assume the measurements from joint encoders are corrupted by white Gaussian noise: $\tilde{\alpha}_t = \alpha + w_t^\alpha$, $\quad w_t^\alpha \sim \mathcal{N}(0_{3,1}, \Sigma^\alpha \delta(t - t'))$. The assumption that the contact point is fixed in the world frame during a contact event leads to $h_p(\tilde{\alpha}_t) = R_t^\mathsf{T}(d_t - p_t) + J_p(\tilde{\alpha}_t)w_t^\alpha$, where $J_p(\tilde{\alpha}_t)$ is the analytical Jacobian of the forward kinematics function.

## 4  Contact Estimation

In this section, we will discuss our deep learning approach for contact estimation. We define our contact state for each leg $l \in \{RF, LF, RH, LH\}$ as $C = [c_{RF} \quad c_{LF} \quad c_{RH} \quad c_{LH}]$ and $c_l \in \{0, 1\}$. Here, 0 denotes no contact, and 1 indicates a firm contact with the ground. Depending on the robot's motion, the contact state vector $C$ can have totally 16 different states. (From all feet are in the air to all feet are in contact with the ground). We formulate it as a classification problem in our deep neural network and aim at estimating the correct contact state $C$ given the input data. In

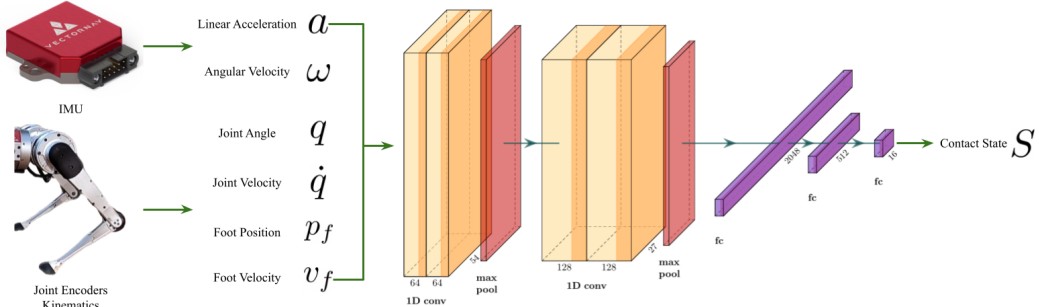

Figure 2: The architecture of the contact estimation network. The network takes in measurements from encoders and an IMU and outputs the contact state of the robot. The structure consists of 2 blocks and 3 fully connected layers. In each block, there are 2 one-dimensional convolution layers and 1 one-dimensional max pooling. The second convolution layer in each block is applied with a dropout mechanism to prevent overfitting.

order to follow typical classification pipelines, we map our contact state vector $C$ to 16 different states $S \in \{0, 1, \ldots, 15\}$ by treating $C$ as a binary value with 4 digits and using binary to decimal conversion as the function. For example, contact state $C_i = \begin{bmatrix} 0 & 1 & 1 & 0 \end{bmatrix}$ is mapped to $S_i = 6$.

## 4.1 Preprocessing of the Input Data

The contact estimation network takes sensor measurements from joint encoders, IMU, and kinematics as input. For a synchronized time $n$, the sensor measurements are concatenated as a $54 \times 1$ array $z_n = \begin{bmatrix} q_n & \dot{q}_n & a_n & \omega_n & p_{fn} & v_{fn} \end{bmatrix}$, where

$$q_n = \begin{bmatrix} q_{RF1n} & q_{RF2n} & q_{RF3n} & q_{LF1n} & \cdots & q_{LH3n} \end{bmatrix}$$
$$\dot{q}_n = \begin{bmatrix} \dot{q}_{RF1n} & \dot{q}_{RF2n} & \dot{q}_{RF3n} & \dot{q}_{LF1n} & \cdots & \dot{q}_{LH3n} \end{bmatrix}$$
$$a_n = \begin{bmatrix} a_{xn} & a_{yn} & a_{zn} \end{bmatrix}, \omega_n = \begin{bmatrix} \omega_{xn} & \omega_{yn} & \omega_{zn} \end{bmatrix}$$
$$p_{fn} = \begin{bmatrix} p_{RFxn} & p_{RFyn} & p_{RFzn} & p_{LFxn} & \cdots & p_{LHzn} \end{bmatrix}$$
$$v_{fn} = \begin{bmatrix} v_{RFxn} & v_{RFyn} & v_{RFzn} & v_{LFxn} & \cdots & v_{LHzn} \end{bmatrix}.$$

Here, $q_n^\mathsf{T}$ is a $12 \times 1$ vector containing all the joint encoder measurements (rad) at time $n$, $\dot{q}_n^\mathsf{T}$ is a $12 \times 1$ vector with joint angular velocity (rad/sec), $a_n$ holds the linear accelerations (m/sec) from the IMU in the IMU frame, $\omega_n$ contains the angular velocity (rad/sec) in the IMU frame, $p_{fn}^\mathsf{T}$ is a $12 \times 1$ vector with foot positions calculated from forward kinematics, and $v_{fn}^\mathsf{T}$ is a $12 \times 1$ vector that carries the linear velocities of each foot. It is worth noticing that both $p_{fn}$ and $v_{fn}$ are represented in the robot's hip frame. To infer the relationship between data across the time domain, for data point at time $n$, we create a window with size $w$ and append previous measurements within this window into a 2D array $D_n = \begin{bmatrix} z_{n-w}^\mathsf{T} & z_{n-w+1}^\mathsf{T} & \cdots & z_n^\mathsf{T} \end{bmatrix}^\mathsf{T}$. $D_n$ is a $w \times 54$ array. (In our case, we use $w = 150$.) Each time, the network takes $D_n$ as input and estimates the contact state $S_n$ as output.

## 4.2 Network Architecture

The contact estimation network consists of 2 blocks of convolutions and 3 fully connected layers, as shown in Figure 2. Each block contains 2 one-dimensional convolution layers and a one-dimensional max pooling. The convolution layers are designed to extract deep features from the input data. We choose a one-dimensional kernel to increase computational efficiency in terms of memory usage and run time. The kernel moves along the time domain, and the data is padded to preserve dimensions. ReLU is employed as the nonlinear activation function for the convolution layer. The second convolution layer is applied with a dropout mechanism to prevent overfitting. At the end of each block, a one-dimensional max-pooling layer is added to downsample the data.

The second block is connected to 3 fully connected layers that convert the deep features into the 16 classes we defined earlier. We also employ the dropout mechanisms in the first 2 fully connected layers to prevent overfitting. Finally, we formulate a classification problem using the cross-entropy loss as $L(P_i) = -\log \frac{\exp(P_i)}{\sum_j \exp(P_j)}$. $P_j$ is the probability output from the network of state $j$, and $P_i$ is the probability of the ground truth state. The detailed network architecture is listed in the Appendix.

Table 1: Number of data of each terrain in the contact data sets.

| | | | | | Terrain Type | | | | | |
|---|---|---|---|---|---|---|---|---|---|---|
| overall | air trotting | air pronking | asphalt road | concrete | forest | grass | middle pebble | small pebble | rock road | sidewalk |
| 1,013,441 | 44,386 | 48,972 | 94,615 | 465,144 | 72,144 | 103,392 | 44,442 | 52,669 | 45,819 | 58,115 |

(a) Robot Setup for data collection.

(b) Different ground types in the contact data set.

Figure 3: (a) Robot configuration for all data sets except grass. Additional RGB-D images are recorded using an Intel D455 camera mounted on the robot. The images are used in ORB SLAM2 to generate ground truth trajectories. (b) Different ground types in the contact data set.

## 4.3   Contact Data Set

We create open-source contact data sets using an MIT Mini Cheetah robot. We record all the proprioceptive sensor measurements from the robot as LCM logs. Recorded measurements include joint encoders data, foot positions and velocities, IMU measurements, and estimated joint torques from an MIT controller [29]. The IMU measurements are received at $1000\,\text{Hz}$, while other data are recorded at $500\,\text{Hz}$. All data are upsampled to match the IMU frequency after being recorded. Around 1,000,000 data points are collected from various terrains, including asphalt road, concrete, forest, grass, middle pebbles, small pebbles, rock road, and sidewalk. To give the network negative examples of non-contact scenarios, we also record several sequences of the robot walking in the air by holding the robot up and giving the same controller command. (i.e., not having contact with the ground while operating the same gait.)

The grass data sets are collected inside an outdoor facility equipped with a motion capture system, which we use as a proxy for ground truth trajectory. Markers are attached to each foot and the robot body for the motion capture system to record the absolute foot positions and body pose in the world frame. For the rest of the data sets, in addition to the proprioceptive sensors, we also record RGB-D images with an Intel D455 camera mounted on top of the robot. We use these images in a state-of-the-art visual SLAM system, ORB SLAM2 [30], to generate the approximate ground truth trajectories. Figure 3b shows the picture of each terrain. The concrete data sets are recorded in a lab environment with polished concrete. The forest data sets are collected with random leaves, woods, and plants scattered around the ground. The middle pebbles data sets are recorded on pebbles ranging from 3 to 7 cm. The small pebbles data sets are recorded on pebbles ranging from 0.5 to 2 cm. Rock road consists of dirt and rocks of random sizes. The contact data sets contain data from 3 different gaits and different robot loading.

The self-supervised network uses an offline algorithm that automatically generates ground truth labels using robot foot height in the hip frame. As shown in the Appendix, the algorithm applies a low-pass filter to the signal and extracts local minima and maxima using the future and past data points around the current timestamp. Hence, it cannot run online on the robot. The contact events are labeled by connecting the local minima between peaks. Moreover, we observe a bouncing effect on the robot's foot upon touch down after inspecting slow-motion videos of Mini Cheetah's walking patterns. The bouncing results in a sudden change of foot height in the signal. Applying the low-pass filter to the signal enables the algorithm to remove these false positives from the ground truth.

## 5   Experimental Results

We train and evaluate the contact network discussed in the previous section. Some sequences are reserved for testing. The rest of the data are split into validation, testing, and training sets with the

Table 2: Accuracy comparison against baselines. The proposed method achieves the highest accuracy on both sequences. Although the gait cycle method has an accuracy closer to the proposed method, it does not remove false positives when gait cycle is violated.

| Sequence | Method | % Accuracy | | | | | % False Positive Rate | % False Negative Rate |
| | | Leg RF | Leg LF | Leg RH | Leg LH | Leg Avg | Leg Avg | Leg Avg |
|---|---|---|---|---|---|---|---|---|
| Concrete Short Loop | GRF Thresholding | 73.43 | 70.02 | 71.69 | 70.04 | 71.30 | 37.07 | 13.24 |
| | Gait Cycle | 85.66 | 84.98 | 84.68 | 85.11 | 85.11 | 22.95 | **0.00** |
| | Proposed Method | **98.34** | **97.87** | **97.95** | **98.56** | **98.18** | **1.45** | 2.51 |
| Grass Test Sequence | GRF Thresholding | 82.55 | 78.93 | 84.62 | 82.48 | 82.14 | 26.87 | **0.63** |
| | Gait Cycle | 92.41 | 92.38 | 91.04 | 90.55 | 91.59 | 10.95 | 3.53 |
| | Proposed Method | **98.08** | **97.57** | **97.73** | **97.73** | **97.78** | **2.35** | 1.98 |
| Forest Test Sequence | GRF Thresholding | 80.99 | 80.09 | 82.75 | 83.24 | 81.77 | 26.54 | 1.84 |
| | Gait Cycle | 83.03 | 82.56 | 84.44 | 84.28 | 83.58 | 24.71 | **0.08** |
| | Proposed Method | **97.05** | **96.62** | **97.24** | **97.40** | **97.08** | **2.82** | 3.12 |

ratio of 15%, 15%, and 70%, respectively. Table 1 lists the number of data being used from each terrain. A window size of 150 is used to allow the network to infer from the time domain. Each data point is normalized along the time domain to prevent scaling issues. We shuffled the training data to ensure generalizability in the model and reduce overfitting. We set the batch size to 30 and use $10^{-4}$ as the learning rate. The training process took 1.5 hours on an NVIDIA RTX 3090 GPU for 30 epochs. We note that after about seven epochs, the network can converge.

## 5.1 Contact Estimation

We evaluate the performance of the network in two ways. First, 16 class denotes the accuracy in terms of 16 contact states $S$ we defined earlier. This case is a harsher way to evaluate the network since it requires all four legs to be correct simultaneously. We also compute the accuracy of the individual leg by comparing the estimated contact state of each leg with ground truth contacts individually. On average, the trained network achieves 93.88% accuracy in 16 contact states and 97.82% in each leg individually. Detailed accuracy across different terrains is listed in the Appendix. In addition, since false positive in contact estimation is fatal to the state estimator, we compute the False Positive Rate (FPR) and the False Negative Rate (FNR) to obtain insights into the performance. The FPR and the FNR are calculated following the definition $FPR = \frac{FP}{FP+TN}$ and $FNR = \frac{FN}{FN+TP}$, where $FP$ is false positive, $TN$ is true negative, $FN$ is false negative, and $TP$ is true positive. On average, our method holds an FPR of 1.82% and an FNR of 2.88% on the test sets.

## 5.2 Comparison of Contact Detection Methods

This experiment compares the accuracy, FPR, and FNR of the proposed method against other contact detection methods using the Mini Cheetah robot. We implement a model-based approach [12, 13, 14] where the estimated ground reaction force is computed via the general equation of motion with a low-pass filter. A fixed threshold is then set to detect the contact events. Because there is no direct access to the motor current on the robot, we use the torque command from the controller to approximate the actual torque on the actuators. We also obtain the gait cycle command from the MIT controller [29] to serve as a contact estimation method.

Table 2 lists the accuracy, FPR, and FNR of the compared methods on three test sequences. We can see that the proposed method achieves the highest accuracy across all sequences and the GRF thresholding has the worst performance. Although the baselines obtain slightly lower FNRs, our method maintains significantly lower FPRs, which is crucial to state estimation tasks.

## 5.3 Contact-Aided Invariant Extended Kalman Filter

We use the contact-aided invariant extended Kalman filter described in Section 3 on concrete test sets. The estimated contacts are converted into LCM messages along with IMU and encoder measurements. The InEKF takes in the LCM messages and estimates the pose of the robot accordingly. We also run the filter using the ground truth contact data to serve as a reference.

Figure 4 shows the trajectories from the InEKF using different contact sources on a concrete loop sequence. Qualitatively compared to the baseline contact detectors, the resulting trajectory with the proposed contact estimation has smaller drifts from the trajectory with ground truth contacts, especially in the height (Y) axis. Furthermore, compared to the baseline contact estimators, the proposed method also yields a smoother trajectory.

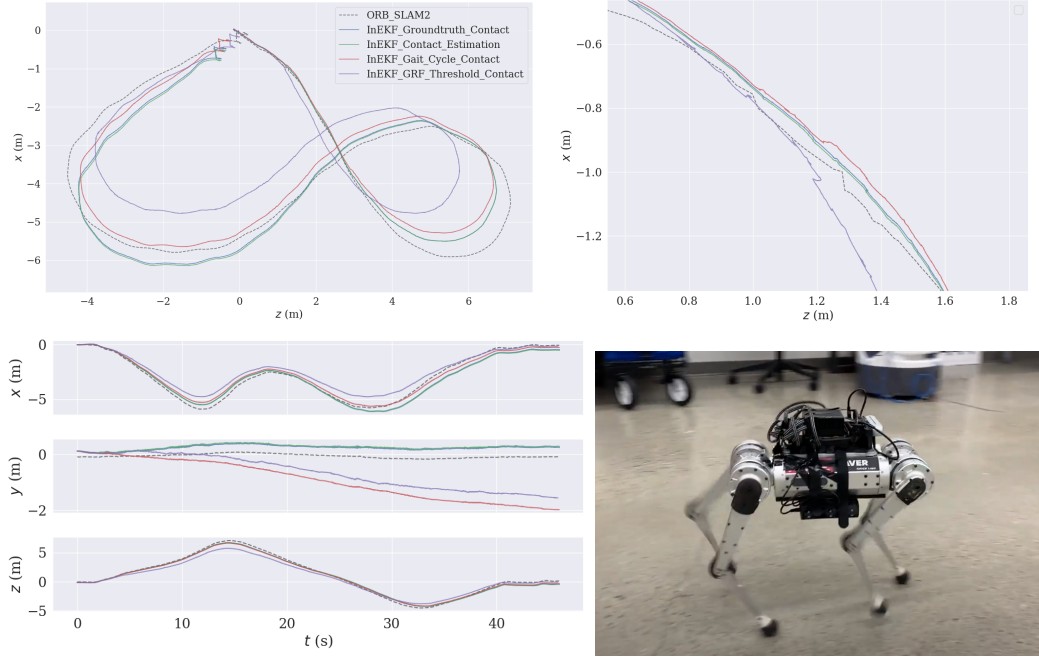

Figure 4: Concrete short loop test sequence. Top Left: The bird's-eye view of the trajectories. The estimated trajectory is mapped to the camera frame ($Y$ pointing downward, and $Z$ pointing forward). Using the estimated contacts, the InEKF can reproduce a similar trajectory to that used ground truth contacts. Top Right: Zoomed-in of the bird's-eye view. The baseline contact estimation methods yield zig-zag trajectories, while the proposed method generates a smoother trajectory. Bottom Left: This figure shows that the gait cycle and GRF thresholding methods produce a significant height (Y) drift in the InEKF state estimator. Bottom Right: Robot configuration in this experiment.

Table 3: The accuracy of the network trained with data from a Cassie-series robot. The network is trained with labels generated from spring deflections and labels from kinematics separately. The network trained with kinematics labels maintains low FPR.

| Contact Labels | % Accuracy | | | | % False Positive Rate | | | % False Negative Rate | | |
|---|---|---|---|---|---|---|---|---|---|---|
| | 4 class | Leg L | Leg R | Leg Avg | Leg L | Leg R | Leg Avg | Leg L | Leg R | Leg Avg |
| Spring Deflection | 98.03 | 99.82 | 98.19 | 99.01 | 0.13 | 2.18 | 1.18 | 0.23 | 1.36 | 0.78 |
| Kinematics | 85.14 | 93.05 | 92.09 | 92.57 | 0.01 | 2.42 | 1.24 | 14.56 | 14.55 | 14.55 |

## 5.4 Cassie-Series Robot Experiment

We also performed an experiment on a Cassie-series bipedal robot, developed by Agility Robotics, to show the applicability of the proposed framework to different robots and training label sources. The robot is equipped with a spring on each leg, which serves as a contact detection sensor. The contact state is defined as $C = [c_L \quad c_R]$. The decimal contact states becomes $S \in \{0, 1, \ldots, 3\}$. Similar to MiniCheetah, we concatenate the input data as $z_n = [q_n \quad \dot{q}_n \quad a_n \quad \omega_n \quad p_{fn} \quad v_{fn}]$. Because data from Cassie is recorded at a higher frequency, a window size of 600 is used.

The data is recorded on polished concrete using a controller developed by Gong and Grizzle [31, 32]. Around $1,300,000$ data points are recorded. Two sequences are reserved as test sets, and two are used in training. We obtained two sources of ground truth contacts in this experiment - one from spring deflections; the other from kinematics. Spring contact labels are obtained by thresholding on the GRF computed from the spring deflection. We assume this is a more reliable contact source as it comes from a physical sensor. The kinematics contacts are generated following the same procedure described in Section 4.3. We train the network using two ground truth contacts separately and evaluate them against the spring contacts.

Table 3 lists the accuracy, the FPR, and the FNR of the two networks. We can see that when trained with ground truth contacts using spring deflection, the network can achieve above $98\%$ accuracy. When trained with the labels generated using kinematics, it can maintain above $92\%$ accuracy on both legs. The results indicate that training using labels generated from the process described in Section 4.3 is useful in practice, albeit with some loss of accuracy. Moreover, the network trained with kinematics labels does not increase the FPR, a valuable trait for state estimation tasks. This experi-

Table 4: The average performance of different network structures on the test sets. Each "block" consists of 2 convolution layers followed by ReLU activation function and 1 max pooling, as described in Figure 2.

| Network Structure | % Accuracy | | | | | | % False Positive Rate | % False Negative Rate |
|---|---|---|---|---|---|---|---|---|
| | 16 class | Leg RF | Leg LF | Leg RH | Leg LH | Leg Avg | Leg Avg | Leg Avg |
| 2 Blocks | **93.88** | **97.76** | **97.66** | **97.86** | **98.00** | **97.82** | **1.82** | **2.88** |
| 1 Block | 93.58 | 97.59 | 97.47 | 97.77 | 97.80 | 97.66 | 1.83 | 3.34 |
| 4 Blocks | 92.19 | 96.90 | 96.64 | 97.03 | 97.56 | 97.04 | 2.28 | 4.28 |
| Conv-Pool-Conv-Pool | 93.13 | 97.45 | 97.19 | 97.46 | 97.46 | 97.39 | 2.06 | 3.67 |

ment also shows the generalizability of the proposed framework. We demonstrate that the proposed framework can work on two different robots as long as one collects enough data for training.

## 5.5 Ablation Study and Runtime

We perform an ablation study on different network architectures to study how the performance changes. Table 4 lists the average performance of different network structure on the test sets. Each "block" consists of 2 convolution layers followed by ReLU activation function and 1 max pooling, as described in Figure 2. In the table, 2 Blocks is the proposed network detailed in Section 4. 1 Block refers to the network with only the first block and the fully connected layers of the proposed network. As for 4 Blocks, we added two additional blocks with 256 and 512 channels before the fully connected layers. Conv-Pool-Conv-Pool consists of only 2 convolution layers of sizes 64 and 128 and the fully connected layers. Each convolution layer is followed by a ReLU function and one max pooling. From the table, we can see that 2 Blocks has the best performance among all other structures. However, 1 Block maintains a smaller network while showing a slightly lower accuracy.

The inference speed on an NVIDIA RTX 3090 GPU is approximately 1100 Hz. The inference speed on an NVIDIA Jetson AGX Xavier, which we equipped on the robot, is around 830 Hz after TensorRT optimization. The frequency of the encoder measurements on Mini Cheetah is 500 Hz. This means the proposed contact estimator run at real-time without delay on the robot. The InEKF runs at 2000 Hz on an Intel i7-8750H CPU.

## 6 Discussion and Limitations

The developed contact estimator can capture contact events for a quadruped robot without force/contact sensors. However, the contacts are modeled as binary values, and the "quality" of contacts is not modeled in our contact estimation network. As a result, a perfect contact is always assumed when a contact event is detected. Furthermore, the covariance matrix of the contact measurements is set heuristically in the InEKF, which along with the uncertainties in the kinematic models, could be the potential sources of estimation error in the InEKF. In the future, we wish to expand the contact detection network to have a covariance estimation of the current contact state. This approach will allow the state estimator to treat contact information as a sensor measurement with an online uncertainty update and potentially decrease the drift in the state estimation.

We formulated the problem as a classification problem with 16 different contact states. This model can potentially be further improved by employing a multi-task learning approach [33, 34]. This approach allows to independently segment the contact events for each foot.

## 7 Conclusion

We developed a multi-modal deep learning-based contact estimation method that does not require contact/force sensors and works well with different robot gaits on distinct terrains. We present open-source contact data sets with self-supervised labels of contact events using an MIT Mini Cheetah robot. We show that the trained contact network can achieve 97% accuracy across different terrains. The estimated contacts are employed in a contact-aided invariant state estimator for quadruped robots, and the resulting trajectory is comparable to a modern visual SLAM system.

## Acknowledgment

Toyota Research Institute provided funds to support this work. Funding for M. Ghaffari was in part provided by NSF Award No. 2118818. This work was also supported by MIT Biomimetic Robotics Lab and NAVER LABS. NVIDIA Corporation provided hardware support for this work.

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
