# OpenReview forum: "Legged Robot State Estimation using Invariant Kalman Filtering and Learned Contact Events"
_robot-learning.org/CoRL/2021/Conference — CoRL2021 Poster_

### Official Review · Reviewer_TWUQ · 2021-07-15

**Originality:** Good
**Technical Quality:** Fair
**Clarity Of Presentation:** Fair
**Impact:** 2

**Recommendation:**

Weak Reject: I recommend rejecting the paper, but will not argue for my recommendation if the majority of other reviewers have a different opinion.

**Summary:**

The paper addresses the problem of legged robot state estimation. Since joint angles can be directly measured using encoders, the central challenge of this problem has been the *global* pose and velocity of the robot (root). Previous work “InEKF” [5] built an extended Kalman Filter incorporating IMU and binary contact sensor measurements to address this problem, utilizing insights such as stance leg being static during contact. And this paper adopts the same method.

The contribution of this paper, is that, since some robots do not have binary contact sensors as the Cassie robot in [5], a neural-network contact classifier is therefore built to predict this info. As standard network architecture and training procedure seem to be good enough for this purpose, the key challenge here is thus how to obtain ground truth contact label in the training data.

For contact labeling, the paper mostly identify contact points by calculating minima height values of feet from motion cycles kinematics. In addition, a bit of manual data cleaning is also needed. (L208)

**Issues:**

Discussed above -- why learning, as well as using gait kinematics to generate ground-truth labels, is justified when we already know the gait pattern.

**Reviewer Expertise:**

Good: General knowledge of the area

**Strengths And Weaknesses:**

Strength:
The idea of generating contact labels offline when we have future info of the trajectory and use them to train an online network for contact prediction is interesting.

Weakness:
The main weakness, in my opinion, is also about the contact prediction network and whether we really need that. Since the Ground Truth contact labels are basically generated by calculating the lowest point of each gait cycle from joint kinematics (L208-209), what the network essentially does is to memorize the kinematics of its own gait.

This raises the immediate question of can we use some simple heuristic rule to replace the network? After all the network will not work when the gait pattern changes, unless we collect new data using the new controller (which seems not easy). But since that would imply we roughly already know the gait pattern beforehand, in my mind heuristic rules would be very easy to implement.

Minor points:
The introduction can be improved by emphasizing what is hard about state estimation in the beginning. I only realized the key is root trajectory estimation (not estimating all joint q’s, they are given) when I skimmed the paper and saw figure 4. Previous work [5] did a better job in clearing that out at first.

The whole section 3 seems to come from [5] — the only change seems to be different robot morphology. It can be made more concise as well.

**Summary Of Recommendation:**

Building such a contact prediction network does not seem difficult if we went thought the data collection. But, as discussed before, I am not sure if machine learning is justified here if we already have the controller ready, and I would love the authors to take some serious re-thinking or let me know what I am missing fundamentally. Again, we should keep in mind that training such a network requires us to already have a working controller, and neural networks in general would not "extrapolate" to other motions/controllers without special treatment.

---

> ### Author Response · Authors · 2021-08-26
> **Respond to Reviewer TWUQ**
>
> **Q1. The main weakness, in my opinion, is also about the contact prediction network and whether we really need that. Since the Ground Truth contact labels are basically generated by calculating the lowest point of each gait cycle from joint kinematics (L208-209), what the network essentially does is to memorize the kinematics of its own gait.**
>
> **A1.** That is an interesting way to look at the network. However, this approach allows us to combine multi-modal signals from IMU, encoders, and kinematic information. We can also collect data using a variety of behaviors and include it in the data set. We emphasize that there is no claim regarding learning the contact dynamics here as it implies solving the causal learning problem (an open problem in machine learning).
>
> **Q2. This raises the immediate question of can we use some simple heuristic rule to replace the network? After all the network will not work when the gait pattern changes, unless we collect new data using the new controller (which seems not easy). But since that would imply we roughly already know the gait pattern beforehand, in my mind heuristic rules would be very easy to implement.**
>
> **A2.** Unfortunately, this is not possible. If we can, it implies we can access a reasonably accurate ground reaction force estimation to do so. If one thinks of the thresholding process as a linear or piecewise linear process, the neural network is exactly doing that. The neural network is a piecewise model (linear layers followed by nonlinear activations, i.e., max function). However, on different ground types, using different gaits and with different robot loading, the robot will need a different set of thresholds, which will be cumbersome to design. The learning-based approach here is doing so automatically using data.
>
> **Q3. The introduction can be improved by emphasizing what is hard about state estimation in the beginning. I only realized the key is root trajectory estimation (not estimating all joint q's, they are given) when I skimmed the paper and saw figure 4. Previous work [5] did a better job in clearing that out at first.**
>
> **A3.** We revised the Introduction by adding the following text after defining the state estimation problem in L. 19: "While joint angles can be directly measured using encoders, the robot's pose and velocity require additional measurements and a mathematically sound data fusion method."
>
> **Q4. The whole section 3 seems to come from [5] — the only change seems to be different robot morphology. It can be made more concise as well.**
>
> **A4.** Thanks for pointing this out. We will reduce the background section and only keep the definitions for introducing the notation.
>
> **Q5. Building such a contact prediction network does not seem difficult if we went thought the data collection. But, as discussed before, I am not sure if machine learning is justified here if we already have the controller ready, and I would love the authors to take some serious re-thinking or let me know what I am missing fundamentally. Again, we should keep in mind that training such a network requires us to already have a working controller, and neural networks in general would not "extrapolate" to other motions/controllers without special treatment. Why learning, as well as using gait kinematics to generate ground-truth labels, is justified when we already know the gait pattern.**
>
> **A5.** The current MIT controller on the robot uses assumed gait cycles (contact events). The gait cycle does not show the contact interval and foot bouncing due to the robot's interaction with the environment. As such, there is a need for contact sensors or some form of computational models (model-based or learning-based). Because the Mini Cheetah robot is lightweight and legs are also very light, we did not get satisfactory results from the robot dynamics model (Lagrangian dynamics). We are currently processing results using this model-based approach to include it in the revised paper as a baseline. In fact, this question was the main motivation for this work.

---

> > ### Comment · Reviewer_TWUQ · 2021-08-31
> > **Reply**
> >
> > I much appreciate the authors' effort, especially the added experiments in sec. 5.2. I thank the authors' honesty in showing that non-learning baselines yield a similar birds-eye view performance for localization (Fig. 4a). (The 2m drifting in height in Fig. 4b on a flat ground does not really make intuitive sense to me and should be easy to fix with simple heuristics/rules.)
> >
> > I am still concerned about the benefit vs added effort of using learning over model-based baselines. Also, the authors mention in the revision that the ground-truth is generated using both past and future info but the network needs to predict using only past info. I think the main reason the network is able to do so is because it is gait style specific (i.e. memorizing certain gait cycle). That makes me worry about the scalability of the method, as other reviewers also mentioned.
> >
> > As such, I change my rating to "weak reject".

---

> > > ### Author Response · Authors · 2021-09-01
> > > **Response to Reviewer TWUQ**
> > >
> > > We thank the reviewer for keeping an open mind and changing the rating after seeing our new efforts. It is indeed interesting to see the birds-eye view of all trajectories. Unfortunately, we did not include the following figure that shows the difference locally (baselines produce rough and noisy trajectories due to false positives) https://i.imgur.com/MsohVtg.png.
> > > Of course, we understand that we missed the opportunity to include it in the supplementary file.
> > >
> > > We agree on the memorization point (as opposed to learning the cause and principles from data). This is the main drawback of current machine learning algorithms. Perhaps we can consider the proposed method useful and certainly not complete. The developed filter provides accurate odometry for a specific robot with known gaits, and in our ongoing research, the same network enables terrain classification and terrain-specific covariance tuning for the filter.

---

> > > > ### Author Response · Authors · 2021-09-04
> > > > **Reference or hint on simple heuristics/rules**
> > > >
> > > > We would highly appreciate it if the reviewer can kindly provide a reference or a pointer on the suggested heuristics/rules for fixing the height drift. Please note that the rules require to be generalizable across different terrains and height variations similar to the state estimator developed here.

---

### Official Review · Reviewer_Ax1o · 2021-07-23

**Originality:** Fair
**Technical Quality:** Good
**Clarity Of Presentation:** Good
**Impact:** 3

**Recommendation:**

Weak Accept: I recommend accepting the paper, but will not argue for my recommendation if the majority of other reviewers have a different opinion.

**Summary:**

The paper addresses contact estimation of a quadruped robot that is not equipped with force/torque sensors on its feet. It fuses the inertial and kinematic motion data of the robot over a time window by a deep convolutional neural network, which outputs a set of binary values (contact/non-contact) with respect to each foot. The training data was collected in several outdoor environments with a help of a motion capture system, and is provided as Open-source quadruped contact data sets recorded using an MIT Mini Cheetah. Using the proposed method, the invariant extended Kalman filter for localization was built and evaluated. It showed a competitive performance with ORB SLAM2.


**Issues:**

To investigate if the trained contact detector works for robots other than the MIT Cheetah by using, for example, a physics simulator, if possible.


**Reviewer Expertise:**

Very good: Comprehensive knowledge of the area

**Strengths And Weaknesses:**

It is remarkable that the same-level accuracy with the state-of-the-art SLAM technique was achieved only with proprioceptive information even in challenging environments. It certainly shows the efficacy of the proposed method.
On the other hand, it seems to be arguable that the method requires huge amount of data recorded over many frames only for detecting if feet are in contact or not. Since the computation resource that a robot can carry is still limited, the availability of the method is a bit questionable for the reviewer. A question which naturally arises in him/her is if it is possible to estimate the position and orientation of the robot without explicitly estimating the contact state. This idea seems preferable as it exploits the computation resource more than the contact detection and might be a shortcut for the purpose. It is probably related with the "quality" of contacts as the authors note.

The reviewer's bigger question is about the generality of the detector, namely, whether the trained detector works for robots other than the MIT Cheetah. Although he/she guesses it does to some extent at least for quadruped robots, he/she wants to know if it is invariant with respect to size, weights and body proportions.


**Summary Of Recommendation:**

I recommend accepting the paper, but will not argue for my recommendation if the majority of other reviewers have a different opinion.

---

> ### Author Response · Authors · 2021-08-26
> **Respond to Reviewer Ax1o**
>
> **Q1. On the other hand, it seems to be arguable that the method requires huge amount of data recorded over many frames only for detecting if feet are in contact or not.**
>
> **A1.** Neural networks, in general, require a large number of examples. But unlike image processing, our signals are low-dimensional and high frequency. It is relatively easy to collect a large amount of data in a  few runs.
>
> **Q2. Since the computation resource that a robot can carry is still limited, the availability of the method is a bit questionable for the reviewer.**
>
> **A2.** The network is lightweight. We have exported the neural network model using Nvidia Tensor RT for real-time deployment. The required speed for a Mini Cheetah is 500 Hz. Currently, the network can be run at 800 Hz on an Nvidia Jetson AGX Xavier. We will include the runtime information in the revised paper.
>
> **Q3. A question which naturally arises in him/her is if it is possible to estimate the position and orientation of the robot without explicitly estimating the contact state. This idea seems preferable as it exploits the computation resource more than the contact detection and might be a shortcut for the purpose. It is probably related with the "quality" of contacts as the authors note.**
>
> **A3.** This is an interesting question because it implies if the contact estimation is an intermediate problem or not. Indeed, the contact estimation itself is not the primary objective. We do not know the answer. But the state of the art high-frequency observers for legged robots use binary contact data explicitly. The answer to this question needs an independent study on its own. We hope our work motivates such investigations.
>
> **Q4. The reviewer's bigger question is about the generality of the detector, namely, whether the trained detector works for robots other than the MIT Cheetah. Although he/she guesses it does to some extent at least for quadruped robots, he/she wants to know if it is invariant with respect to size, weights and body proportions. To investigate if the trained contact detector works for robots other than the MIT Cheetah by using, for example, a physics simulator, if possible.**
>
> **A4.** We are including experiments on a Cassie-series biped robot developed by Agility Robotics. We would like to point out that training the network on one robot and testing it on another with different specifications will not work. However, we also suspect if a large data set will be available, pre-training is an option, analogous to image processing tasks.

---

> > ### Comment · Reviewer_Ax1o · 2021-09-03
> > **Thank the authors for their response.**
> >
> > The reviewer thanks the authors for their response.
> > The answer A2 makes sense.
> > The reviewer agrees A3 that it is another issue, although any discussion about this in the revised manuscript is welcome.
> > A4 is not surprising for the reviewer, and thus, he/she does not have any comments on it.
> > He/she is doubtful about A3 since the training data should be collected in various situations and the quantity of the data set itself does not make sense.
> > To conclude, the reviewer does not change the evaluation.

---

### Official Review · Reviewer_RPT4 · 2021-07-23

**Originality:** Fair
**Technical Quality:** Good
**Clarity Of Presentation:** Very Good
**Impact:** 2

**Recommendation:**

Weak Reject: I recommend rejecting the paper, but will not argue for my recommendation if the majority of other reviewers have a different opinion.

**Summary:**

This paper proposes a supervised deep learning method for contact detection for quadrupedal robots, a critical subroutine for many state estimation and control methods for legged robots. The authors integrate their learned detector in one such state estimator, which is shown to perform nearly as well as ground-truth contact state.

**Issues:**

Resolved Major Issues:
1) Authors should either conduct a comparative study with other contact detection methods, or convincingly argue why the ones discussed in the paper are not suitable for comparison.

Remaining Major Issues:
2) Authors should conduct a control experiment with the contact detector and state estimator in the control loop.


Nitpick:

There appears to be some issue in the calculation of RMSE of the relative pose error. In the Appendix, the “Sidewalk+Asphalt Road” example is claimed to only have 3 [cm] of error per frame, yet the provided figure (Fig. 4) seems to imply that the estimation is off by nearly 20 [m] by the end of the trajectory.


**Reviewer Expertise:**

Good: General knowledge of the area

**Strengths And Weaknesses:**

Strengths:
1) The performance of the method is particularly impressive given the limited sensing modalities (no contact sensors or parallel/series spring deflections) that it leverages.
2) Not only was the detection approach clearly explained, but also the authors provided a great high-level summary of the invariant EKF used in the paper.

Resolved Weaknesses:
1) The authors do not provide a performance comparison to any of the other cited contact detection methods.

Weaknesses:
1) A critically important characteristic of any state estimator is that it does not excite instability when used in real-time as the input to a controller. This may be of particular issue in this approach if the associated controller requires the contact state. However, the authors only use the method to reconstruct trajectories offline, which does not provide insight into in-the-loop performance.



**Summary Of Recommendation:**

The authors propose a great concept for learned contact estimation in the absence of contact sensing. However, without in-the-loop experiments or a comparative study, it is difficult to assess the suitability or performance of the method for state estimation. Furthermore, the reason for needing the learned module for localization and mapping tasks is not sufficiently motivated in the text. Thus the paper does not seem ready for publication.

---

> ### Author Response · Authors · 2021-08-26
> **Respond to Reviewer RPT4**
>
> **Q1. Authors should either conduct a comparative study with other contact detection methods, or convincingly argue why the ones discussed in the paper are not suitable for comparison.**
>
> **A1.** We are currently processing results for model-based contact detection in the experiments. Unfortunately, we cannot find open-source libraries for this work, and we are implementing it ourselves.
>
> **Q2. The authors propose a great concept for learned contact estimation in the absence of contact sensing. However, without in-the-loop experiments or a comparative study, it is difficult to assess the suitability or performance of the method for state estimation. As this appears to be the central purpose of the proposed work, the paper does not seem ready for publication. Authors should conduct a control experiment with the contact detector and state estimator in the control loop.**
>
> **A2.** Currently, the controller only receives the estimated body velocity as feedback and relies on preplanned gait cycles. In addition, the controller focuses on information about the beginning of a contact event. The observer requires the reliable duration that the robot foot is in contact with the ground. We are working on a real-time library for the developed network and observer; however, it requires more software engineering and is part of our future work. We acknowledge the important research question about the performance of the controller using estimated contact events. We conjecture that since the learning-based method uses real robot-environment data to estimate the contact event, in theory, it should inform the controller better than a fixed preplanned cycle. This is an interesting future study and somewhat orthogonal to the goal of this paper. The odometry estimates for perception are highly valuable for localization and mapping tasks.
>
> **Q3. There appears to be some issue in the calculation of RMSE of the relative pose error. In the Appendix, the “Sidewalk+Asphalt Road” example is claimed to only have 3 [cm] of error per frame, yet the provided figure (Fig. 4) seems to imply that the estimation is off by nearly 20 [m] by the end of the trajectory.**
>
> **A3.** The metric is Relative Pose Error (RPE), introduced in Section 5.2 (L. 253). The large error on the figure will show the absolute trajectory error that is a metric for SLAM systems. For dead reckoning/odometry systems, RPE is often used as the system will necessarily drift over time. These metrics are standard and, for example, are available in the TUM RGB-D SLAM data set (https://vision.in.tum.de/data/datasets/rgbd-dataset/tools#evaluation).

---

> > ### Comment · Reviewer_RPT4 · 2021-08-27
> > **Response to Authors**
> >
> > Thank you to the authors for taking the time to address each of these concerns. I have addressed the comments & I look forward to seeing the revised paper / supplement when it is uploaded.
> >
> > **Q1/A1:** A comparison against the model based methods will be a great addition to the paper.
> >
> > **Q2/A2:** After reading this comment, I do now realize that the method in fact is useful for localization & mapping, and that the experiments offer compelling evidence to that end. I will however point out that the Introduction section in the paper heavily focuses on the usefulness of this observer for real-time control. The first paragraph denotes "*state estimation*," not localization & mapping, as the primary goal of concern for the paper, and particularly highlights that >1kHz estimation "is crucial for controller and planners to maintain stability and execute planned policies." In short, I still think a real-time in-the-loop experiment is critical to the impact of the paper as long as the project is portrayed in this way.
> >
> > **Q3/A3:** I see from your comment that I misunderstood the table; thank you for the correction. I agree that RPE is the right metric to use here and that the number in the paper looks reasonable.

---

> > > ### Author Response · Authors · 2021-08-31
> > > **Response to Reviewer RPT4**
> > >
> > > Thank you for following up with our comment.
> > >
> > > We have added comparisons against a model-based method and against the contacts obtained from gait cycle.
> > >
> > > Thank you for your constructive suggestion. We modified the introduction to better emphasize localization and mapping. Due to the short period of time, we were unable to incorporate the proposed framework into a controller. However, in the paper, we demonstrate the effectiveness of the proposed method in the aspect of localization and mapping, which is the main focus of this work.

---

> > > > ### Comment · Reviewer_RPT4 · 2021-09-04
> > > > **Final Comments**
> > > >
> > > > Thank you to the reviewers for their diligence in responding to my concerns.
> > > >
> > > > In the end, I do not feel the localization & mapping application is compelling enough for publication at CoRL.
> > > >
> > > > It seems to me what makes the model-based methods struggle with misidentification is the fact that they are causal--an important feature for real-time control, but not for localization & mapping. By contrast, your data processing analyzes the entire trajectory to find local minima, and I haven't read anything in the rebuttal or paper which makes believes that this cannot be done entirely programmatically. Therefore it is not clear to me why such a computationally intensive learning pipeline is necessary. I also agree with TWUQ that the y-axis drift can likely be eliminated by something simple.
> > > >
> > > > I do however think that this method could possible be interesting for real-time control. I do hope the authors continue this work and integrate the estimator with real-time control, in which case I think the paper will have significantly more impact. My recommendation will remain at "weak reject."

---

> > > > > ### Author Response · Authors · 2021-09-04
> > > > > **Response to Reviewer RPT4**
> > > > >
> > > > > Thank you for also sharing your final comments. We respectfully disagree that localization and mapping are not compelling topics as they are advertised officially by the organizers here https://www.robot-learning.org/author-information/call-for-papers
> > > > >
> > > > > We followed the call for paper and our work fits into:
> > > > > - State estimation, localization and mapping
> > > > > - Multimodal perception, sensor fusion, and computer vision
> > > > >
> > > > > The reviewer is right that causal inference might not be necessary for localization and mapping tasks, unlike real-time control. At this time, we are not studying the control problem, in fact, we assume a controller is given and will not be developed further. Therefore, we find it slightly unfair to be penalized for lack of contribution that we do not intend to have.
> > > > >
> > > > > For data collection, even though the process is partly automated, we use future and past information for labeling. Therefore, it is not possible to solve the problem entirely programmatically using kinematic information. In fact, if we do, this approach will fall back on our model-based baseline. Besides, deep learning frameworks use data to learn representations that work better as more data becomes available. This motivation cannot be satisfied using a rule-based program (the opposite approach of deep learning).
> > > > >
> > > > > We would highly appreciate it if the reviewer can kindly provide a reference or a pointer on the suggested heuristics/rules for fixing the height drift. Please note that the rules require to be generalizable across different terrains and height variations similar to the state estimator developed here.

---

> > > > > > ### Comment · Reviewer_RPT4 · 2021-09-04
> > > > > > **Clarificaiton**
> > > > > >
> > > > > > I apologize for the lack of clarity on this comment and I will try to restate better. Again, I first want to reiterate that I think the approach may be interesting for use as a real-time estimator that feeds into a real-time controller, and that I hope future work will show this to be true.
> > > > > >
> > > > > > I did not mean that localization & mapping is an uninteresting application space; I just do not feel that this project makes a compelling contribution to localization & mapping. Primarily this comes from the fact that the majority of the paper centers on a deep-learned approach, which I find well-motivated for real-time estimation in a control loop (which you have not applied it to), but poorly motivated for localization & mapping (which is your choice of application).
> > > > > >
> > > > > > The reason I find the approach compelling for real-time control is that it would allow you to predict contact detection online, which is learned from data generated offline by your non-causal approach. This exact fact is highlighted in your [other response to review aydR](https://openreview.net/forum?id=yt3tDB67lc5&noteId=mdg3_haw0eW).
> > > > > >
> > > > > > I want to clarify that in the case of localization & mapping, I believe you can use the entire trajectory in a non-causal way to estimate the contact sequence, and then input that into your estimator. In this case, you use the "future and past" data for reading between the current time and t = 0, just as your processing script does. At this point, the primary motivation left for the deep-learning approach is that it can reject false positives, because a human has removed them in post-processing. Specifically to your comment, "Besides, deep learning frameworks use data to learn representations that work better as more data becomes available," supervised learning is only as good as the ground-truth labels you provide, and the question here is if you could just provide the ground-truth directly.
> > > > > >
> > > > > > If this is the only major motivation, then significant convincing is required that the quantity of false-positives that cannot be programmatically removed is meaningful. Unfortunately, I am only able to see 3 provided examples in the paper (Fig. 3c), which could all be accounted for programmatically by rejecting points that don't meet a velocity/Lipschitz threshold.
> > > > > >
> > > > > > I do not have a reference on the z-height drift, and I recognize I could be wrong. I want to clarify that this was not of significant impact to my rating, rather a suggestion to double check your implementation as it seems unusual; I would have looked through the code myself if it was provided.

---

> > > > > > > ### Author Response · Authors · 2021-09-04
> > > > > > > **Follow up**
> > > > > > >
> > > > > > > Thank you for your humble and honest response. We want to point out that we are in complete agreement with the reviewer that our approach for use as a real-time estimator that feeds into a real-time controller is a promising research direction on its own, and future work will show that. But we need to stay true to our motivation and why we did this work. We cannot simply pivot now. Furthermore, we need this odometry on the robot for our future research whether it deserves a conference publication for presenting the idea or not (discussions here show that it can spark interesting conversation!).
> > > > > > >
> > > > > > > Two reasons we stay away from the real-time control experiments are:
> > > > > > > 1. One must **prove** that the deep neural network is a **stable** observer (small changes in the input leads to small changes in the output),
> > > > > > > 2. and **prove** that the network provides sufficient **robustness** (bounded inputs lead to bound outputs).
> > > > > > > Without the above theoretical results (both are **open problems** as far as we know) that a control paper needs to have, the work will be empirical work with little to no value in safety-critical applications.
> > > > > > >
> > > > > > > Our code and data sets are publicly available already, but we cannot share them here (double-blind policy). If the reviewer googles the paper's title, it should not be difficult to find the repositories.

---

### Official Review · Reviewer_aydR · 2021-07-24

**Originality:** Very Good
**Technical Quality:** Very Good
**Clarity Of Presentation:** Fair
**Impact:** 3

**Recommendation:**

Weak Reject: I recommend rejecting the paper, but will not argue for my recommendation if the majority of other reviewers have a different opinion.

**Summary:**

The paper presents a deep learning based approach for estimating contact information from proprioceptive sensory data. First, the authors collected the training data by deploying locomotion controllers on various terrains. The ground-truth data is estimated with the normal component of foot positions in the body frame (line 206-207) or captured with a motion capture system (line 207-208). Then the paper learns neural networks that consist of several convolutional and fully connected layers. Once the contact states are estimated, it can be fed into Contact-Aided Invariant Extended Kalman Filtering to estimate the global location. The authors have demonstrated that the proposed method can estimate the contact status at the accuracy of 80%~95%. And they also have demonstrated that the estimated contacts are helpful for localization.

**Issues:**

My major concerns are 1) data collection and 2) overfitting. Particularly, the data collection issue can make the entire work significantly less impactful. Hope to get some clarification during the rebuttal phase.

**Reviewer Expertise:**

Very good: Comprehensive knowledge of the area

**Strengths And Weaknesses:**

+ The paper applies a machine learning technique for state estimation.
+ The proposed method is validated in two folds: local level (contact state estimation) and global level (path reconstruction).
- The ground-truth data needs some clarification
- The learned model may be overfitted to the robot and controller.


**Summary Of Recommendation:**

I am positive about this paper, but I would save my full support because of some drawbacks. I would like to see the authors’ responses before making the final decision. Until then, my support is lukewarm.

The first confusion is about the ground-truth contact information. The paper seems to say that they are *estimated* from the foot height in the body frame (except for the grass scenario). Then why do we need to estimate it using machine learning? We can simply apply the same procedure to estimate the ground-truth contact information from the foot height. This issue is particularly odd because this can nullify all the motivation of the paper. And this is also connected to the evaluation. The results presented in Table 3 are also compared against the *estimated* ground-truth? If yes, I don’t think it is a fair comparison.

My other concern is about overfitting to the motion. The given dataset seems to be generated with a fixed controller (model-based controller? deep RL controller? maybe better to describe more details). However, we are not sure whether the proposed method can be generalized to different motions or gaits . For instance, can we use the same model for bounding gaits, or manually created jumping motions? I would suggest the authors discuss this generalization issue in the text.

As stated in the paper, I am not sure we can easily assume perfect contacts without foot slips. I guess it happens quite often during data collection, particularly on certain terrains such as pebbles or grass. This would limit the impact of the proposed work. But it seems not clear what the authors can do further on top of the current discussion in Section 6.

I am not too familiar with the Contact-Aided Invariant Extended Kalman Filter technique, but the validation with this seems reasonable. It will be better if Figure 4 illustrates the locations where the model fails to predict.

---

> ### Author Response · Authors · 2021-08-26
> **Respond to Reviewer aydR**
>
> **Q1. The first confusion is about the ground-truth contact information. The paper seems to say that they are estimated from the foot height in the body frame (except for the grass scenario). Then why do we need to estimate it using machine learning? We can simply apply the same procedure to estimate the ground-truth contact information from the foot height. This issue is particularly odd because this can nullify all the motivation of the paper. And this is also connected to the evaluation.**
>
> **A1.** The initial ground truth contact labels come from the foot height in the body frame. However, this process will not produce correct data sets. Based on a set of explainable rules, we manually clean up the data to generate the final data sets. Therefore, it is not possible to generate contact data automatically online using this method, and human supervision is required. The main reason that foot heights do not produce correct contact labels directly is that the foot bounces off the ground after each contact, leading to a large number of false positives.
>
> **Q2. The results presented in Table 3 are also compared against the estimated ground-truth? If yes, I don’t think it is a fair comparison.**
>
> **A2.** We agree that the estimated contact ground truth is not perfect. However, we carefully clean data to ensure there are no false positives (when the foot is not in contact and the label says contact) in data. The false negatives will not severely impact the observer. We are running an experiment using a Cassie-series robot to compare the quality of our estimated ground truth contacts against measured contact data using spring deflections. We will include the new experiment in the revised paper.
>
> **Q3. My other concern is about overfitting to the motion. The given dataset seems to be generated with a fixed controller (model-based controller? deep RL controller? maybe better to describe more details).**
>
> **A3.** We currently run the default MIT controller that is based on an MPC (model-based). The controller is developed by the robot developers and provides several gait patterns. We will cite the appropriate reference for this in the revised paper.
>
> **Q4. However, we are not sure whether the proposed method can be generalized to different motions or gaits . For instance, can we use the same model for bounding gaits, or manually created jumping motions? I would suggest the authors discuss this generalization issue in the text.**
>
> **A4.** Yes, as long as data related to that gait or behavior/jump is included in the training data. Our current data set includes several jumping motions.
>
> **Q5. As stated in the paper, I am not sure we can easily assume perfect contacts without foot slips. I guess it happens quite often during data collection, particularly on certain terrains such as pebbles or grass. This would limit the impact of the proposed work. But it seems not clear what the authors can do further on top of the current discussion in Section 6.**
>
> **A5.** The reviewer is right that the proposed method will not assess the quality of contact. In fact, we are working on online covariance estimation using the same neural network to provide a quality metric. However, it is not ready yet. Furthermore, the Invariant EKF in its current derivation requires binary contact measurements. The foot slip up to some degree can be accommodated by tuning the covariance of the contact-inertial process. However, we are not addressing extreme cases in this work as the goal is to provide a state estimator for legged robots without contact sensors comparable to those with such dedicated hardware.
>
> **Q6. I am not too familiar with the Contact-Aided Invariant Extended Kalman Filter technique, but the validation with this seems reasonable. It will be better if Figure 4 illustrates the locations where the model fails to predict.**
>
> **A6.** Thanks for pointing this out. That is exactly why we need to use the Contact-Aided Invariant Extended Kalman Filter as a proxy to know if the contact estimation by the network is useful or not. If the filter does not work as expected then clearly the network is not usable. The filter is sensitive to false positives, and the performance will be degraded. In terms of complete failure, since the network only relies on proprioceptive signals, unless the computer fails, it will always be available.

---

> > ### Comment · Reviewer_aydR · 2021-08-28
> > **Additional Comments**
> >
> > I thank the authors for thorough responses.
> >
> > **Data processing**
> > Then the authors must provide more details of "manual" processing. Would you be able to describe the procedure? What kind of prior knowledge/principle is needed?  Can this method be scalable? I am worrying about the possibility where the entire method relies on "manual" ground-truth generation.
> >
> > **Overfitting**
> > My concern is that the learned contact estimator is specific to the given controller and cannot be generalized to other controllers. Then the learned controller would be not so useful. I would appreciate it if the authors can give us more explanation about this.

---

> > > ### Author Response · Authors · 2021-08-31
> > > **Response to Reviewer aydR**
> > >
> > > Thank you for following up with our responses.
> > >
> > > **Data Processing** In the revised paper, we list the algorithm being used in pre-processing the ground truth contacts and describe the principle for manually removing the false positives. An example figure of the signal is also included in the paper. We also conduct an experiment on a Cassie-series bipedal robot. The Cassie robot is equipped with a spring sensor on each leg, which is assumed to be a more reliable source for contact detection. We train the network with ground truth labels generated with the process described in Section 4.3 and compared it against the contacts from spring deflection. The results show only a small loss of accuracy.
> > >
> > > However, the authors would like to emphasize the failure-prevent aspect of the proposed framework:
> > > For robots with contact sensors, the contacts from the sensors can be easily collected to train the network. In the case of contact sensor failures, the trained contact detector can take over and maintain enough odometry accuracy. The experiment on the Cassie robot is a good example of this.
> > >
> > > **Overfitting** Although the proposed method is datasets-dependent, the experiment indicates that as long as the gait patterns are included in the training sequence, the network is able to successfully detect contact with different gaits. From Cassie's experiments, we show that with several rounds of data collection, the proposed method can be easily deployed on other robots.

---

> > > > ### Comment · Reviewer_aydR · 2021-09-02
> > > > **Adjustment**
> > > >
> > > > My concerns are not well addressed in the rebuttal, so I lowered my rating.
> > > >
> > > > **Data processing** So the impact of the work seems to be weak, in the sense that the data processing should not be fully automated (if yes, we can use that automated algorithm) nor manual (in this case, learning highly depends on human effort). The current manuscript is currently taking a weird balance between them.
> > > >
> > > > **Overfitting** I am not mentioning other robots. The proposed method can be applied to any robot, as the authors successfully demonstrated. It is more about controllers. But the generated data seems to be overfitted to the given controller. Can you use it for developing other controllers? I don't see any evidence.

---

> > > > > ### Author Response · Authors · 2021-09-04
> > > > > **Response to Reviewer aydR**
> > > > >
> > > > > We thank the reviewer for additional comments and the intellectual conversion about our work. Please see our additional response to the remaining concerns below.
> > > > >
> > > > > **Data processing.**  A fully automated data processing is called *self-supervised* learning. Self-supervised learning is a subset of unsupervised learning methods. Fully manual labeling is called supervised learning. Semi-supervised learning is also possible if data is partially labeled.
> > > > >
> > > > > Our approach is consistent with the current industry standards. The method is supervised. However, the initial labels (containing false positives) are generated automatically. This is a desired feature rather than being negative. A human supervisor removes the false positives. Therefore, we consider the entire pipeline as supervised learning.
> > > > >
> > > > > Another question raised by the reviewer is that even if we can automatically generate labels (self-supervision), why not running the same labeling algorithm online? The answer is that we use future and past information for labeling. Therefore, it is not possible to solve the problem entirely programmatically using kinematic information. In fact, if we do, this approach will fall back on our model-based baseline. Besides, deep learning frameworks use data to learn representations that work better as more data becomes available. This motivation cannot be satisfied using a rule-based program (the opposite approach of deep learning). Moreover, the rule-based algorithm will only be able to solve one problem, whereas the representation learning approach can be extended to multi-task networks, e.g., for terrain classification, covariance estimation, and potentially other applications.
> > > > >
> > > > > **Overfitting.**  The reviewer is right that the network is overfitted to a given controller. At this time, we are not studying the control problem, in fact, we assume a controller is given and will not be developed further. Our method solves the intended problem, i.e., providing reliable odometry. Therefore, we find it slightly unfair to be penalized for lack of contribution that we do not intend to have.

---

> > > > > > ### Comment · Reviewer_aydR · 2021-09-06
> > > > > > **Response to the authors**
> > > > > >
> > > > > > Thanks for the author's detailed response. Although my decision remains the same, I would like to provide more constructive feedback upon the authors' respectful comments.
> > > > > >
> > > > > > **Data processing.** SL can be useful when we do not have a clear algorithm to solve a problem. Or label themselves can be vague. But here, I got the impression that the problem looks possible to be solved by a mathematical algorithm. Therefore, this is very different from typical SL problems. Yes, learning may get rid of the dependency on future variables (while sacrificing the accuracy), but still, a prescribed algorithm plays the most important role for the entire pipeline. And the algorithm is even not perfect: it still requires manual supervision. Manual supervision implies bad reproducibility, in general.
> > > > > >
> > > > > > **Overfitting.** The author may want to revise the paper significantly if want to stress the contributions to localization. Particularly, consider changing the title and overview figure.
> > > > > >
> > > > > > **Conclusion.** I am not too negative about the work itself. Just needs some improvement.  My above two comments are mostly about revising the story of the paper and redefining the contributions. Looking forward to seeing the published manuscript soon.
> > > > > >
> > > > > > p.s. I am not too sure about the clear definition of the "industry standard".

---

### Author Response · Authors · 2021-09-12
**Update on Automatic Labeling and Summary of Our Response**

We want to thank all reviewers again. The provided feedback and conversions have been extremely helpful in understanding and framing the problem better and developing the initial idea into a more useful method. We could not ask for a better outcome from the peer-review process, and we will certainly submit our work to CoRL again.

We can now successfully generate contact labels automatically (self-supervised training). We used a very strong low-pass filter to find modes of foot position signals and separate contact intervals. This approach works, of course, only offline for data set generation. Although there are some false positives compared to human-labeled data, the state estimation performance seems to be equally good on the polished concrete data set. Please see the following results (figures):

https://ibb.co/XSKymPn

https://ibb.co/ZgDLp5Z

https://ibb.co/YycgPkG

https://ibb.co/Jpb12S1

As far as we have searched, existing works on legged robot state estimation that relate to this work are also gait-specific. However, we do not think this is a problem for perception, as any robot will ultimately have a fixed set of behaviors. Furthermore, our vision is that robotics algorithms will be robot-specific. There will be no ultimate robot-agnostic software stack that works on any robot (as much as we find this realization somewhat painful). For example, the notation of traversability can be very different for different robots even within the same category of mechanisms.

Fixing the height drift using generalizable heuristics on all tested terrains in the wild is non-trivial (at least for us). Therefore, we would like to push back on the comment that the provided performance is not novel wrt existing methods. We welcome specific references and suggestions on this matter, and otherwise, this is a significant contribution among the class of state estimators for legged robots.

We can now run the state estimator in real-time on the robot (not part of this work but shows that it works and will be publicly available on a GitHub repository). Finally, we agree with the comments on choosing a better title and writing a better story in the introduction. We believe that is a minor change and can be done easily.

---

### Meta-Review · Area_Chair_FneZ · 2021-08-16

**Recommendation:** Accept (Poster)
**Confidence:** 5

**Metareview:**

The authors consider a deep-learning based approach to estimate contacts for legged robots.  Physical contact sensors on legged robots typically fail due to repeated impacts and the proposed contact estimator using multi-modal proprioceptive sensory data is important and critically needed.  The authors should clarify how the ground-truth contact information is obtained - if no additional sensors were used, then why use this learning-based approach?  The authors should also discuss what fraction of their data has feet slipping.  Please provide more information on the computation time / induced delays in using this estimator.  This will be important in using the estimator stably in a feedback loop.  Finally, any thoughts on what part of the data is robot specific? Can the collected data be used to estimate contact on other legged robots?

I have taken the author responses and discussion with the reviewers into consideration. I commend the authors on making a sincere attempt to address all the reviewer comments and feedback.  I think accurate contact detection is a critical component of legged robot research.  The authors provide information in Table in Sec. IV showing the improved performance using their method.  I am recommending an acceptance with poster.

It will be interesting to find out ratio of false positives vs false negatives - I presume one of this is more important for control.

---

> ### Author Response · Authors · 2021-08-26
> **Response to Area Chair FneZ**
>
>
> Thank you for the comprehensive reviews and constructive feedback. A summary of the changes is given below. A point-by-point response to each of the reviewers' suggestions is posted directly below. The questions are directly taken from the feedback.
>
>
> Summary of changes for the revised paper:
> * Running a new experiment using a Cassie-series biped robot to compare the quality of our estimated ground truth contacts against measured contact data using spring deflections.
> * This experiment will also test the proposed method on another legged robot.
> * Processing results for model-based contact detection in the experiments as a baseline.
> * Including more information and discussion on runtimes.
> * Clarifying the data set creation process in the revised paper to ensure it will be reproducible.
> * Adding Invariant EKF results using contacts from gait cycle commands. These contact events are preplanned assumptions.
>
> **Q1. The authors should clarify how the ground-truth contact information is obtained - if no additional sensors were used, then why use this learning-based approach?**
>
> **A1.** The initial ground truth contact labels come from the foot height in the body frame. However, this process will not produce correct data sets. Based on a set of explainable rules, we manually clean up the data to generate the final data sets. Therefore, it is not possible to generate contact data automatically online using this method, and human supervision is required. The main reason that foot heights do not produce correct contact labels directly is that the foot bounces off the ground after each contact, leading to a large number of false positives.
>
> **Q2. The authors should also discuss what fraction of their data has feet slipping.**
>
> **A2.** We are running an experiment using a Cassie-series robot to compare the quality of our estimated ground truth contacts against measured contact data using spring deflections. For Mini Cheetah, currently, we cannot know this information. As the goal is to have the state estimator running onboard, so long as the pose and velocity estimates are correct, the system can tolerate the error in contact estimation.
>
> **Q3. Please provide more information on the computation time / induced delays in using this estimator. This will be important in using the estimator stably in a feedback loop.**
>
> **A3.** We have exported the neural network model using Nvidia Tensor RT for real-time deployment. The required speed for a Mini Cheetah is 500 Hz. Currently, the network can be run at 800 Hz on an Nvidia Jetson AGX Xavier. We will include the runtime information in the revised paper.
>
> **Q4. Finally, any thoughts on what part of the data is robot specific? Can the collected data be used to estimate contact on other legged robots?**
>
> **A4.** The data set and the trained networks are robot specific. However, the method can be easily deployed on other legged robots so long as we collect data. We have verified this on a Cassie-series biped robot and will include the experiments in the revised paper or supplementary file.

---

> > ### Comment · Reviewer_TWUQ · 2021-08-31
> > **Second aydR's point**
> >
> > I second aydR's point. The paper seem to be placing itself in an awkward position in terms of ground-truth labeling. If it claims the training data generation is mostly automatic with programmable rules, then one may argue we do not need to train this network but instead can use those rules at run time. If it claims the training data labeling is mostly manual, then the fundamental concern would be the feasibility of doing so for each different robot x each different motion/task.

---

> > > ### Author Response · Authors · 2021-08-31
> > > **Response to Area Chair FneZ**
> > >
> > > Thank you for following up with our comment.
> > >
> > > In the revised paper, we list the algorithm being used in pre-processing the ground truth contacts. The pre-processing algorithm computes the local minimum and maximum value using the data points both in the future and past around the current timestamp. We do not have measurements for future signals when running in real-time. Thus, the algorithm can only run in an offline manner.
> > >
> > > In the revised paper, We also describe the principle for manually removing the false positives. An example figure of the signal is also included in the paper. By inspecting the slow-motion videos of Mini Cheetah's walking patterns, we observe a bouncing effect on the robot’s foot upon touch down. This results in a sudden change of foot height. Because of this, we manually remove labels where a sudden change of the foot height takes place.
> > >
> > > We also conduct an experiment on a Cassie-series bipedal robot to validate the ground truth generation process. The Cassie robot is equipped with a spring sensor on each leg, which is assumed to be a more reliable source for contact detection. We train the network with ground truth labels generated with the process described in Section 4.3 and compared it against the contacts from spring deflection. The results show only a small loss of accuracy.
> > >
> > > However, the authors would like to emphasize the failure-prevent aspect of the proposed framework: For robots with contact sensors, the contacts from the sensors can be easily collected to train the network. In the case of contact sensor failures, the trained contact detector can take over and maintain enough odometry accuracy. The experiment on the Cassie robot is a good example of this.

---

### Decision · Program_Chairs · 2021-09-13

**Decision:**

Accept (Poster)

**Comment:**

The authors consider a deep-learning based approach to estimate contacts for legged robots.  Physical contact sensors on legged robots typically fail due to repeated impacts and the proposed contact estimator using multi-modal proprioceptive sensory data is important and critically needed.  The authors should clarify how the ground-truth contact information is obtained - if no additional sensors were used, then why use this learning-based approach?  The authors should also discuss what fraction of their data has feet slipping.  Please provide more information on the computation time / induced delays in using this estimator.  This will be important in using the estimator stably in a feedback loop.  Finally, any thoughts on what part of the data is robot specific? Can the collected data be used to estimate contact on other legged robots?

I have taken the author responses and discussion with the reviewers into consideration. I commend the authors on making a sincere attempt to address all the reviewer comments and feedback.  I think accurate contact detection is a critical component of legged robot research.  The authors provide information in Table in Sec. IV showing the improved performance using their method.  I am recommending an acceptance with poster.

It will be interesting to find out ratio of false positives vs false negatives - I presume one of this is more important for control.